# Vertex model with internal dissipation enables sustained flows

Jan Rozman [1,4], KVS Chaithanya [2,3,4], Julia M. Yeomans [1] ✉ & Rastko Sknepnek [2,3] ✉

Complex tissue flows in epithelia are driven by intra- and inter-cellular processes that generate, maintain, and coordinate mechanical forces. There has been growing evidence that cell shape anisotropy, manifested as nematic order, plays an important role in this process. Here we extend an active nematic vertex model by replacing substrate friction with internal viscous dissipation, dominant in epithelia not supported by a substrate or the extracellular matrix, which are found in many early-stage embryos. When coupled to cell shape anisotropy, the internal viscous dissipation allows for long-range velocity correlations and thus enables the spontaneous emergence of flows with a large degree of spatiotemporal organisation. We demonstrate sustained flow in epithelial sheets confined to a channel, providing a link between the cell-level vertex model of tissue dynamics and continuum active nematics, whose behaviour in a channel is theoretically understood and experimentally realisable. Our findings also show a simple mechanism that could account for collective cell migration correlated over distances large compared to the cell size, as observed during morphogenesis.

Collective cell migration, i.e. a coordinated movement of groups of cells that maintain contacts and coordinate intercellular signalling[1,2], underlies wound healing[3,4], cancer invasion[5,6], and embryonic development[7,8]. Proper execution of these biological processes necessitates the generation and robust spatiotemporal regulation of large-scale coordinated flows[9]. How tissue-scale flows emerge from coordination between molecular processes and cell-level behaviours such as cell intercalation, division, and ingression is poorly understood.

In vitro experiments on epithelial cell monolayers grown on soft substrates[10–13] reveal that collective cell migration is an emergent active phenomenon that requires maintenance, transmission, and coordination of mechanical forces over distances large compared to the cell size. Dense cell packing and fluctuations driven by active cellular processes lead to correlated patterns in cell displacements[14,15] reminiscent of the glass transition between liquid and solid phase. This transition is related to cell shape[16–18] and cell-cell adhesion[19,20], allowing epithelia to control their rheological properties.

Cell monolayers are, therefore, a prime example of an active system and theories of the physics of active matter[21–24] have been instrumental in understanding many aspects of their collective behaviour[25–27]. In particular, particle-based approaches to cells self-propelled by locally generated directed forces (i.e. polar forces) have been used to understand collective cell motion[12,28,29]. However, by Newton's third law, cells can only self-propel against an external structure, e.g. a substrate. At the same time, large-scale collective migration can also occur without a solid substrate, for example, during early-stage embryonic development such as gastrulation in avian embryos[30]. The absence of a substrate excludes polar self-propulsion, suggesting that the process must be driven by cells actively pulling and pushing against each other. As these forces are internal to the tissue, the leading contribution must be dipolar, resembling active nematics[22,31]. Such dipolar forces can arise, e.g. due to tension in cell-cell junctions[32–38] or cell-level active nematic stresses[39,40]. Moreover, as there is no friction with the substrate, dissipation can only be internal

[1]Rudolf Peierls Centre for Theoretical Physics, University of Oxford, Oxford, UK. [2]School of Life Sciences, University of Dundee, Dundee, UK. [3]School of Science and Engineering, University of Dundee, Dundee, UK. [4]These authors contributed equally: Jan Rozman, KVS Chaithanya. ✉e-mail: julia.yeomans@physics.ox.ac.uk; r.sknepnek@dundee.ac.uk

to the tissue. It is, therefore, important to understand how collective motion can emerge without self-propulsion and dissipation through substrate friction.

While without a substrate, polar active forces and substrate friction are absent, this does not imply the reverse. The dynamics of supported epithelia can still depend on internal dissipation[19,41] and resemble active nematics[13,42]. Therefore, while nematic activity and internal dissipation are necessary to understand flows in unsupported epithelia, their effects are of broad relevance to the general understanding of epithelial mechanics.

Vertex models are a widely-used class of models for understanding tissue dynamics[43–45]. They encode the geometry of each cell in a confluent epithelium and can capture physics inaccessible to particle-based approaches, such as a zero-temperature rigidity transition[46,47]. However, in almost all cases, dissipation is assumed to be proportional to the local vertex velocity, i.e. cells locally exchange momentum with the substrate and the momentum of the cell layer is not conserved. In the context of active matter physics, such systems are referred to as being in the dry limit[21]. However, without a substrate, the dissipation must solely be internal to the tissue[48,49], so that total momentum is conserved, placing unsupported epithelia into the class of wet active matter systems[21] (see in ref. 21 for a review of wet and dry active matter models). The properties of wet active vertex models are not well understood.

Using a channel geometry combined with cell-level nematic activity, we show that the vertex model with internal dissipation in an active fluid phase develops spontaneous directional flows due to long-range velocity-velocity correlations, which we do not observe in the substrate dissipation case. This model, therefore, provides the plausible necessary ingredients for a cell-level description of large-scale active flows in epithelia not supported by a substrate. It also outlines a possible connection between cell-level and continuum descriptions based on theories of active nematics[50] that can capture morphogenic flows[51,52], but where the activity is introduced phenomenologically. In addition to being relevant for modelling early development, geometric constraints have been shown to impact collective cell migration, e.g. by a width-driven transition from a state of no net flow to shear flow in elongated retinal pigment epithelial cells and mouse myoblasts[53], or geometry-controlled global and multinodal oscillations in Madin-Darby canine kidney (MDCK) cells[54–56].

## Results

We study the vertex model in its canonical form[43] in which the energy function reads

$$E_{\text{VM}} = \sum_c \left[ \frac{K_A}{2} \left( A_c - A_0 \right)^2 + \frac{K_P}{2} \left( P_c - P_0 \right)^2 \right], \quad (1)$$

where the sum is over all cells. $K_A$ and $K_P$ are, respectively, the area and perimeter elasticity moduli, $A_c$ and $P_c$ are the area and perimeter of cell $c$, whereas $A_0$ and $P_0$ are the target area and perimeter, taken to be the same for all cells. The usual approach assumes overdamped dynamics with dissipation modelled as the vertex-substrate frictional force $-\xi \mathbf{v}_i$, where $\xi$ is the friction coefficient and $\mathbf{v}_i$ is the velocity of vertex $i$ (Fig. 1a). This is, however, not appropriate for suspended epithelia and an alternative model for dissipation is necessary, e.g. as recently proposed in ref. 49. The dissipative force on vertex $i$ is instead dominated by contributions arising from the difference between its velocity and the velocities of its neighbours (Fig. 1b). The equation of motion thus becomes

$$\xi \dot{\mathbf{r}}_i + \eta \sum_{j \in \mathcal{S}_i} \left( \dot{\mathbf{r}}_i - \dot{\mathbf{r}}_j \right) = -\nabla_{\mathbf{r}_i} E_{\text{VM}} + \mathbf{f}_i^{\text{act}}, \quad (2)$$

where $\eta$ is the vertex-vertex friction coefficient and the sum is over vertices $j$ in the star $\mathcal{S}_i$ of (i.e. connected to) the vertex $i$. $E_{\text{VM}}$ is the elastic energy, $\nabla_{\mathbf{r}_i}$ is the gradient with respect to the position vector $\mathbf{r}_i$

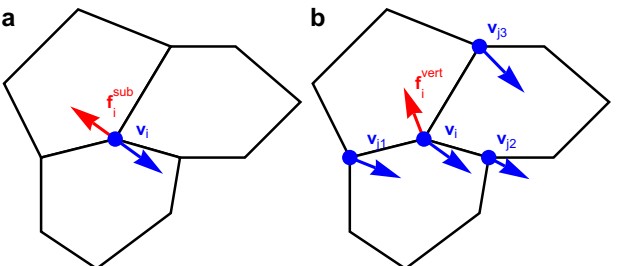

**Fig. 1 | Schematic of different dissipation models. a** In the substrate dissipation model, each vertex experiences a frictional force due to the substrate. This force always points in the exactly opposite direction to the vertex velocity. **b** In the internal dissipation model, friction on a vertex is due to its motion relative to the other vertices it is connected to, and the direction of the friction force therefore also depends on the velocity of those vertices.

of vertex $i$, the overdot indicates the time derivative, and $\mathbf{f}_i^{\text{act}}$ is the active force on vertex $i$. For $\eta = 0$, equation (2) reduces to the familiar overdamped equation of motion, and we refer to this limit as the substrate dissipation model. Inversely, we refer to the $\eta \gg \xi$ case as the internal dissipation model. A term akin to $\dot{\mathbf{r}}_i - \dot{\mathbf{r}}_j$ has been used to model Vicsek-type alignment in several models of self-propelled particles[12,29,57]. The key difference is that the present model does not include self-propulsion and cell-cell alignment. The equations of motion instead describe the low Reynolds number limit of a model with internal dissipation rather than the inertial limit of models with self-propulsion.

The active force on vertex $i$ arises from cell-level stresses that depend on cell elongation (Supplementary Fig. 1)[40]. It is given as $\mathbf{f}_i^{\text{act}} = - \tilde{\mathbf{r}}_i \cdot \boldsymbol{\sigma}_c$, where $\boldsymbol{\sigma}_c = -\zeta \mathbf{Q}_c$ is the stress tensor[31] and $\tilde{\mathbf{r}}_i = \frac{1}{2} (\mathbf{r}_{i+1} - \mathbf{r}_{i-1}) \times \hat{\mathbf{z}}$, with $\mathbf{r}_{i+1}$ and $\mathbf{r}_{i-1}$ being respectively the vertices following and preceding $i$ in counterclockwise order around cell $c$[58,59], and $\hat{\mathbf{z}}$ is the unit-length vector normal to the plane of the cell. $\zeta$ measures strength of activity and $\mathbf{Q}_c = \left( \frac{1}{P_c} \sum_j \ell_j \hat{\mathbf{t}}_j \otimes \hat{\mathbf{t}}_j \right) - \frac{1}{2} \mathbf{I}$ is the cell-shape anisotropy tensor. Here, the sum is over all junctions $j$ of cell $c$, $\ell_j$ is the length of the $j$th junction, $\hat{\mathbf{t}}_j$ is a unit-length vector along said junction, and $\mathbf{I}$ is the identity tensor. For sufficiently high positive values of $\zeta$, this model with substrate dissipation dynamics generates active chaotic flows reminiscent of extensile active nematics in terms of, e.g. the velocity profiles around $\pm 1/2$ topological defects[40]. Here we, therefore, focus on the $\zeta > 0$ (i.e. extensile) case with activities generally in the range where the model tissue behaves as an active nematic fluid (see 'Methods' for details of model implementation). Lastly, we emphasise that the model does not feature active polar (i.e. self-propelled) terms, and the sum of active forces a cell generates is zero.

We choose $A_0 = 1$, $K_A = 1$, and $\eta = 1$ ($\xi = 1$) for the internal (substrate) dissipation model. This sets the units of length as $\sqrt{A_0}$, time as $\eta/(K_A A_0)$ (or $\xi/(K_A A_0)$ for the substrate dissipation model), energy as $K_A A_0^2$, stress as $K_A A_0$, and the shape parameter $p_0 = P_0/\sqrt{A_0} \equiv P_0$. In these units, $E_{\text{VM}} = (1/2) \sum_c [(A_c - 1)^2 + K_P (P_c - p_0)^2]$. Unless stated otherwise, $K_P = 0.02$ and $p_0 = 3.85$ (i.e. slightly above the originally reported rigidity transition threshold for the passive vertex model[46]). In the internal dissipation case, we set $\xi/\eta = 10^{-5}$, and for the substrate dissipation case, $\eta = 0$ so that all dissipation is external, as noted above.

We aim to understand the minimal cell-level ingredients for collective motion to arise from nematic activity. Drawing analogies with studies of active nematics which have analytically, numerically, and experimentally shown system-wide flows when confined to a channel[50,53,60], we study the active nematic vertex model with both internal and substrate dissipation under channel confinement. To create a channel in the vertex model, vertices of cells in the outermost top and bottom layers are affixed to a line, representing no-slip boundary conditions. As cells on the channel wall have their spacing set by the starting hexagonal lattice, but the extensile activity results in

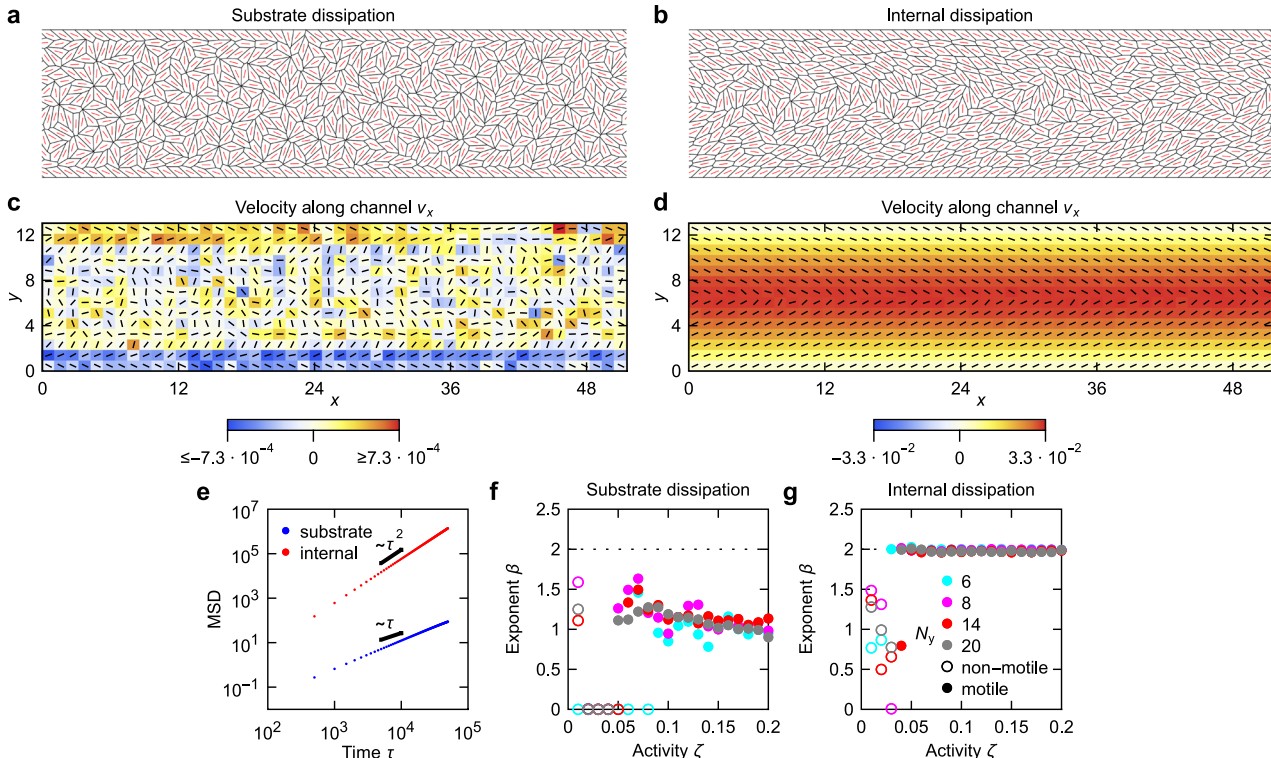

**Fig. 2 | Comparison of substrate and internal dissipation model dynamics in a channel. a, b** Model tissue at $t = 2 \times 10^5$ for a substrate (**a**) and internal (**b**) dissipation model at $\zeta = 0.1$ and size $N_x = 48$ by $N_y = 14$ cells. Cell directors are shown in red. **c, d** The corresponding cell velocities along $x$ for the substrate (**c**) and internal (**d**) dissipation model, averaged between $t = 1 \times 10^5$ and $t = 2 \times 10^5$ in increments of $\delta t = 500$, overlaid with the averaged director profiles (black lines). **e** Mean-squared

displacement as a function of time for the substrate and internal dissipation model tissues from (**a**) and (**b**), respectively. The best power-law fit exponents are 1.2 (substrate dissipation) and 2.0 (internal dissipation). **f, g** Exponent of $at^\beta$ fit to MSD for the substrate dissipation (**f**) and internal dissipation (**g**) model; horizontal dotted line at $\beta = 2$ marks ballistic motion. Values of the prefactor $a$ are inconsequential and not shown. Source data are provided as a Source Data file.

cell elongation, they tilt at an angle to the wall. Periodic boundary conditions are applied along the length of the channel. See 'Methods' for details of channel implementation.

We first simulate a channel of length $N_x = 48$ and width $N_y = 14$ cells using only substrate dissipation dynamics with activity $\zeta = 0.1$, finding no evidence of channel-wide unidirectional flows (Fig. 2a, c and Supplementary Movie 1). By contrast, a model with internal dissipation develops clear unidirectional flows (Fig. 2b, d and Supplementary Movie 2). To quantify flows, we measure the mean-squared displacement of cells after flows emerged, confirming that the cell movement is ballistic in the internal dissipation, but not in the substrate dissipation model (Fig. 2e and 'Methods'). We next scan over activities in the range $\zeta = 0 - 0.2$ for different channel widths. We find that ballistic transport is ubiquitous in the internal dissipation case for sufficiently high activities, but never develops with substrate dissipation (Fig. 2f, g and 'Methods'). Moreover, measuring the angle between cell velocity and the channel direction also shows that unidirectional flows only emerge in the internal dissipation case (Supplementary Fig. 2 and 'Methods').

In Fig. 3a, we plot how the mean cell velocity along the channel with internal dissipation depends on the activity and the channel width ('Methods'), showing that both higher activities and wider channels lead to higher mean velocities. In Fig. 3b we plot the velocity profiles in the channel with internal dissipation as a function of the activity for $N_y = 14$. For sufficiently high $\zeta$ the cell velocities reach a maximum in the centre of the channel and decrease towards the walls, resembling a Poiseuille-like flow, as seen in continuum theories[61–63].

Interestingly, the threshold activity $\zeta_c$ for flows to develop in Fig. 3a does not depend strongly on the channel width. As it is possible that this is because spontaneous flows simply do not develop on the time scale of the simulation, we performed additional simulations that

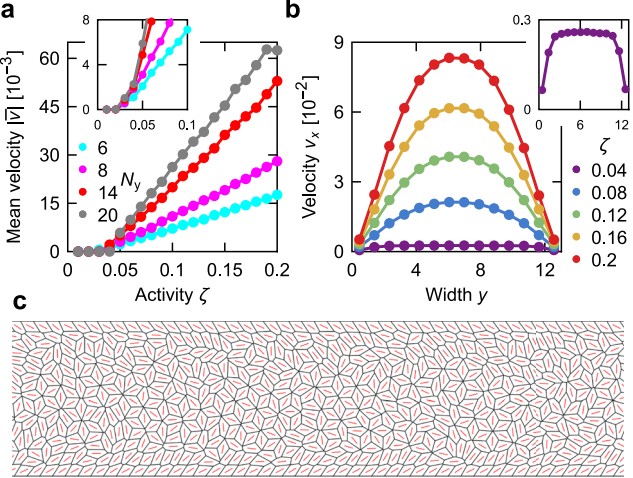

**Fig. 3 | Flows in a channel. a** Mean velocity as a function of activity for different channel widths in the internal dissipation model. The inset shows the region near the threshold for simulations starting from a flow configuration ('Methods'). **b** Velocity profile across the channel width for different activities. The inset shows the $\zeta = 0.04$ profile. **c** Model tissue with $p_0 = 3.85$, $\zeta = 0.04$ at $t = 2 \times 10^5$. Panels (**b**) and (**c**) are for $N_y = 14$. Source data are provided as a Source Data file.

initially start at a higher activity so that flows develop. After that, we instantaneously reduce the activity to the final value and continue the simulation ('Methods'). While this slightly reduces the threshold value, it remains mostly independent of the width of the channel (inset of Fig. 3a). Moreover, except near the transition, mean velocities along

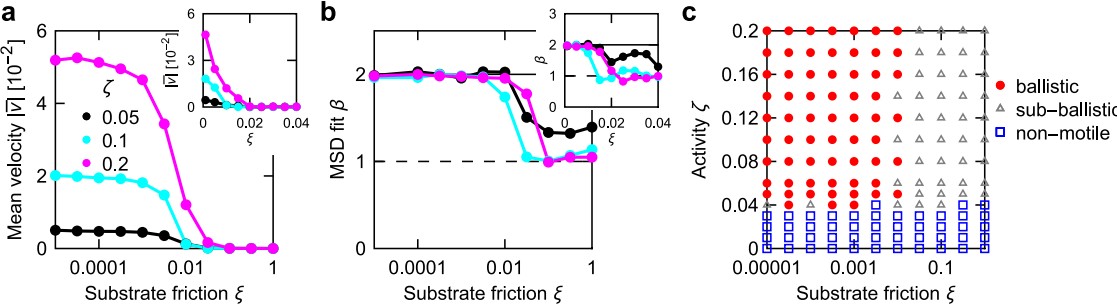

**Fig. 4 | Cell motion transitions from ballistic to sub-ballistic as substrate friction increases. a, b** Mean velocity along the channel (**a**) and exponent of $at^\beta$ fit to MSD (**b**) as a function of substrate friction for different activities. Insets of both panels focus on the transition region using a linear scale. **c** Phase diagram spanning substrate friction and activity. Data on all panels for $N_y = 14$. Points on (**a**, **b**) are connected for clarity. Source data are provided as a Source Data file.

the channel are very similar for both initial conditions (Supplementary Fig. 3).

For a narrow range of activities, i.e. up to ≈0.02 above the threshold critical value required for flows, the flow profile across the channel crosses over to that of plug flow, which is uniform, except close to the boundaries (inset of Fig. 3b). In effect, cells in the bulk move little relative to each other, while still flowing relative to the no-slip cells at the channel wall (Fig. 3c and Supplementary Movie 3). This behaviour is a consequence of the rhombile state[40], where the system no longer resembles active nematics. Therefore, the flow activity threshold predictions of continuum theories[50,61] are not expected to apply to it. Lastly, note that the threshold activity does depend on the target perimeter $p_0$, with higher $p_0$ corresponding to lower $\zeta_c$ (Supplementary Fig. 4). This is likely because the energy cost for T1 transitions decreases (and eventually vanishes) with increasing $p_0$. However, the threshold activity does not decay to 0 for the entire studied range of target perimeters, up to $p_0 = 4$, indicating that the absence of flows at low $\zeta$ is not just due to the passive vertex model being in its solid phase.

We next focus on how substrate friction affects the dynamics of the system at constant internal dissipation $\eta = 1$. We first analyse the $N_y = 14$ channel. As substrate friction $\xi$ increases, the net velocity along the channel decreases (Fig. 4a and 'Methods') while transport transitions from being ballistic to sub-ballistic (Fig. 4b). In Fig. 4c, we present a phase diagram spanning substrate friction and activity, showing a ballistic regime at low and a sub-ballistic regime at high values of friction $\xi$; low activities correspond to non-motile tissues for all values of substrate friction. Supplementary Fig. 5 shows the corresponding phase diagrams for $N_y = 8$ and $N_y = 20$. The phase diagrams for different $N_y$ are qualitatively similar. However, the boundary between ballistic and sub-ballistic motion moves towards lower values of $\xi$ as the width of the channel increases.

For sustained unidirectional flows, the system correlation length has to be comparable with the channel width[60]. In Fig. 5, we plot the velocity-velocity and director-director correlation functions ('Methods') for an $N_x = N_y = 200$ model tissue with periodic boundary conditions for varying values of the vertex-substrate friction $\xi$. For the substrate dissipation model, ($\xi = 1$ and $\eta = 0$), both correlation functions rapidly decay at distances comparable to a single cell. With decreasing $\xi$ while setting $\eta = 1$, the correlation lengths quickly grow, reaching ~30 cell lengths at $\xi = 10^{-5}$ for the velocity-velocity correlation. Therefore, if the tissue is an active fluid, long-range correlations require internal dissipation, which is typically not included in the vertex models. Inversely, substrate friction introduces screening[41,64–66], which reduces the correlations, with higher substrate friction corresponding to a shorter screening length. When substrate friction is sufficiently high, the correlation length becomes too short for unidirectional flows to develop. Rather, cell motion is similar to that of an unconfined system (i.e. resembles active turbulence[40]). This agrees

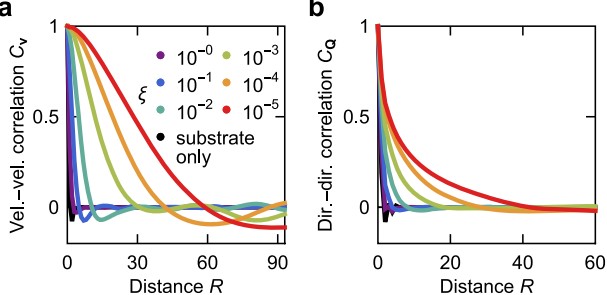

**Fig. 5 | Internal dissipation dynamics allow for longer-range correlations in the vertex model.** Velocity-velocity (**a**) and director-director (**b**) correlation functions at $\zeta = 0.1$ for different values of the vertex-substrate friction $\xi$ at $\eta = 1$, as well as for the substrate dissipation model ($\xi = 1$, $\eta = 0$). Both panels are for a periodic model tissue with $4 \times 10^4$ cells at $t = 2 \times 10^4$. Source data are provided as a Source Data file.

with the boundary between ballistic and sub-ballistic regions moving towards lower substrate friction for increasing channel width in the phase diagrams shown in Supplementary Fig. 5. Wider channels require a higher correlation length for unidirectional flow and, hence, friction to be lower.

We also check whether internal dissipation is required for sustained unidirectional flows regardless of target perimeter $p_0$. Setting $p_0 = 4$, well above the passive rigidity transition reported to be $p_0 \approx 3.81$[46], again shows that sustained unidirectional flows only emerge in the internal dissipation model (Supplementary Fig. 6). In the opposite limit, for $p_0 = 1$, i.e. deep in the solid regime of the passive model, the behaviour resembles continuum active nematics only for activities above a threshold $\zeta_*$ (with $\zeta_* = 0.37$ at $p_0 = 1$), as reported in ref. 40. Above $\zeta_*$, internal dissipation is again required for ballistic transport (Supplementary Fig. 6). For a range of activities below $\zeta_*$, the system is instead in the rhombile state, which has a finite shear modulus[40]. Due to the finite shear modulus, internal dissipation is not required for long-range correlations, and we indeed find that ballistic transport can develop with only substrate friction (Supplementary Fig. 6). The rhombile state, however, does not resemble commonly studied epithelia, so this regime is of limited interest. We also remark that for all studied $p_0$, even if simulations initially include internal dissipation to develop a flow configuration but are then switched to substrate dissipation, sustained unidirectional flows cease above $\zeta_*$ (Supplementary Fig. 7 and 'Methods'). We emphasise that in the case of an active nematic fluid internal dissipation is necessary for sustained unidirectional flows but it is not sufficient. Biological factors such as cell activity, cell-cell adhesion, and junctional tension also control the transition to unidirectional flows (Figs. 2f, g, 3a, and 4; Supplementary Figs. 2, 4–7), in agreement with experimental observations[19,20].

 4

Lastly, to establish that observed directional flows are not affected by the channel length, we perform simulations for $N_x = 200$ (setting $N_y = 8$ or 14 and $\zeta = 0.1$ or 0.2). The resulting velocity profiles are essentially identical to those in the $N_x = 48$ case (Supplementary Fig. 8).

## Discussion

A biologically relevant realisation of a channel would be a differentiated stripe of active cells embedded in otherwise passive tissue that would, e.g. mimic the region of an embryo that is involved in convergent extension[67]. Supplementary Fig. 9 and Supplementary Movie 4 show such an example, with a stripe of softer extensile cells ($p_0 = 3.85$, $\zeta = 0.1$) surrounded by a solid passive tissue ($p_0 = 1$, $\zeta = 0$), with periodic boundary conditions. Again using the internal dissipation model, we find that the differentiated cells start to flow along the stripe. As the system uses internal dissipation and is periodic, the passive bulk cells are pushed in the opposite direction. Given that an external tissue presents a rheologically more complex boundary than a simple wall, it would be interesting to analyse our model in the context of viscoelastic confinement[68].

This study shows that internal dissipation dynamics allow the vertex model to develop channel-wide, unidirectional flows in an active nematic fluid phase which we do not observe in the substrate dissipation model. Moreover, internal dissipation dynamics allows a non-confined model tissue to develop long-range correlations in velocity and nematic order, whereas correlations in the substrate dissipation model are limited to only the scale of a single cell. The short correlation length scale implies that if sustained flows in the substrate dissipation model are at all possible, much greater fine-tuning of parameters would be required for them to develop. On the other hand, in the internal dissipation model, coherent, unidirectional flows are ubiquitous for a wide range of parameters. A recent study using phase-field models similarly observed longer-range correlations arising as a result of cell-cell friction[69]. Finally, while we are focusing on nematic activity here, unidirectional channel flows in cell-level models have also been reported due to self-propulsion forces[70–73] and chiral tension modulations[74,75].

The results highlight the importance of different modes of dissipation for collective active cell migration. Moreover, they show that the standard approach to vertex model dynamics based on substrate friction may be insufficient to explain crucial processes during embryonic development in which long-range flows emerge, e.g. as during the formation of the primitive streak in the chicken embryo[30]. This is especially important in some early-stage embryos where cells are not supported by a substrate. Studying active nematic vertex models in confinement in the presence of internal and substrate dissipation also offers a way to better relate them to the well-analysed continuum active nematics[60]. As vertex models are one of the most common mechanical models of epithelia, and nematic order and activity have recently been shown to play an important role in that type of tissue[13,76–78], a better understanding of how to connect this cell-level approach to the extensive literature on continuum active nematic models is crucial.

## Methods
### Channel model
We study a channel of length $N_x = 48$ cells starting with a hexagonal initial condition, with all cells having their initial areas set to $A_0$. Before starting the simulation, all non-affixed vertices are perturbed in a random direction by a displacement with magnitude drawn from the uniform distribution [0, 0.1] so that cells have a starting elongation.

In simulations, we use a time-step $\Delta t = 0.01$. If a junction length falls below 0.01, a T1 transition is performed, and the length of the new junction is set to 0.011. Vertices on the channel wall are not allowed to move as a proxy for no-slip boundary conditions, and T1 transitions are forbidden on junctions that have a vertex on the wall.

### Solving the equations of motion
Since $x$ and $y$ directions are decoupled, equation (2) can be rewritten as[49]

$$\mathbf{M}\dot{\mathbf{r}}_\gamma = \mathbf{f}_\gamma. \tag{3}$$

For a system with $N$ vertices, $\dot{\mathbf{r}}_\gamma$ and $\mathbf{f}_\gamma$ are $N$−dimensional vectors with components of, respectively, the velocity of and the force on each vertex along $\gamma \in \{x, y\}$, and $\mathbf{M}$ is an $N \times N$ sparse matrix. Assuming that each vertex is connected to three neighbours, the non-zero matrix elements are $M_{ii} = \xi + 3\eta$ and $M_{ij} = -\eta$ for $i \neq j$ if vertices $i$ and $j$ share an edge. For a fixed vertex $k$, $M_{kj} = \delta_{kj}$ and $f_{\gamma,k} = 0$. Using $\dot{\mathbf{r}}_\gamma \approx [\mathbf{r}_\gamma(t + \Delta t) - \mathbf{r}_\gamma(t)]/\Delta t$ and rearranging terms gives

$$\mathbf{M}\,\mathbf{r}_\gamma(t + \Delta t) = \mathbf{M}\,\mathbf{r}_\gamma(t) + \Delta t\mathbf{f}_\gamma \tag{4}$$

or

$$\mathbf{r}_\gamma(t + \Delta t) = \mathbf{r}_\gamma(t) + \Delta t\mathbf{M}^{-1}\mathbf{f}_\gamma. \tag{5}$$

For periodic boundary conditions, $\mathbf{M}$ is singular for $\xi = 0$, and $\xi$ was set to at least $10^{-5}$ in all simulations (including those of channels). Numerically, rather than computing the matrix inverse $\mathbf{M}^{-1}$, it is more efficient to solve equation (4) directly using a sparse linear system solver, e.g. provided by the Eigen3 library[79].

### Velocity and director profiles
To determine the average velocity and director profiles in Fig. 2b, d and Supplementary Fig. 9, the model tissue is divided into an $N_x \times N_y$ grid[40]. The velocity along $x$ of the plaquette $(i, j)$ is

$$v_{x,(i,j)} = \frac{\sum_{c,t} h_{(i,j)}[\mathbf{r}_c(t)] v_{x,c}(t)}{\sum_{c,t} h_{(i,j)}[\mathbf{r}_c(t)]}, \tag{6}$$

where the sum is over all cells $c$ and times between $t = 10^5$ and $t = 2 \times 10^5$ in steps of $\delta t = 500$. $\mathbf{r}_c(t)$ is the centre (mean of vertices) of the cell $c$ at time $t$, $v_{x,c}(t)$ is the $x$ component of the velocity of cell $c$, defined as the average velocity of its non-fixed vertices, and the vertex velocity $\mathbf{v}_i(t) = [\mathbf{r}_i(t) - \mathbf{r}_i(t - \Delta t)]/\Delta t$. Finally, the function $h_{(i,j)}(\mathbf{r})$ takes the value 1 if $\mathbf{r}$ lies within plaquette $(i, j)$, and 0 otherwise.

Similarly, the $\mathbf{Q}$ tensor of the plaquette $(i, j)$ is

$$\mathbf{Q}_{(i,j)} = \frac{\sum_{c,t} h_{(i,j)}[\mathbf{r}_c(t)] \mathbf{Q}_c(t)}{\sum_{c,t} h_{(i,j)}[\mathbf{r}_c(t)]}, \tag{7}$$

with $\mathbf{Q}_c(t)$ being the $\mathbf{Q}$ tensor of cell $c$ at time $t$. The mean velocities along the channel width in Fig. 3b and Supplementary Fig. 8 are obtained in the same way, except that there are now only $N_y$ plaquettes, each representing one row along the channel.

### Mean-squared displacement and mean velocity
To determine the mean-squared displacements shown in Fig. 2e and used for fitting in Figs. 2f, g and 4b, c as well as Supplementary Figs. 5, 6a, b, g, h, and 7a–c, simulations are first run until $t = t_1$ so that flows can emerge. Afterwards, the simulation is continued until $t = t_1 + t_2$, using this period to calculate MSD using

$$\text{MSD}(\tau) = \frac{1}{N'_c N_\tau} \sum_t \sum_c |\mathbf{r}_c(t + \tau) - \mathbf{r}_c(t)|^2, \tag{8}$$

where the first sum is over times $t$ between $t_1$ and $t_1 + t_2$ in intervals $\delta t = 500$ for which $t + \tau \leq t_1 + t_2$, $N_\tau$ is the number of such times $t$, the second sum is over all cells that are not attached to the wall, $N'_c$ is the number of those cells, and the equation takes into account periodic boundary conditions. The nature of the transport is then determined by fitting $at^\beta$ to the MSD function between $\tau = 0$ and $\tau = t_2/2$. If MSD

never exceeds $10^{-6}$, we assume $\beta = 0$, and if MSD$(\tau = t_2/2) < 1$, the simulation is classified as non-motile.

The mean velocities along the channel used in Figs. 3a and 4a as well as in Supplementary Figs. 3, 4, and 8 are determined by comparing the displacement of cells along $x$ between $t_1$ and $t_1 + t_2$, so that the equation reads

$$\bar{v} = \frac{1}{N_c t_2} \sum_c \left[ x_c(t_1 + t_2) - x_c(t_1) \right], \qquad (9)$$

where the sum is over all $N_c$ cells (and the equation takes into account periodic boundary conditions). We set $t_1 = t_2 = 10^5$ for the internal dissipation model and $t_1 = t_2 = 3 \times 10^5$ for the substrate dissipation model (except for Fig. 2e, where $t_1 = t_2 = 10^5$ is used for both internal and substrate dissipation).

When analysing channels starting from a flow configuration (inset of Fig. 3a and Supplementary Figs. 3 and 7), the simulation first runs with internal dissipation for $t = t_0$ with activity $\zeta_0$ so that the flow configuration can develop. They then instantaneously switch to the final activity and dissipation mode and run until $t = t_0 + t_1$ before measurements begin. The measurements are performed until $t = t_0 + t_1 + t_2$. The expression for MSD [equation (8)] is corrected accordingly. We set $t_0 = t_1 = t_2 = 10^5$; $\zeta_0 = 0.45$ for $p_0 = 1$ and otherwise $\zeta_0 = 0.1$ for $p_0 > 3.8$ and $\zeta_0 = 0.2$ for $p_0 \leq 3.8$.

### Velocity angle quantification

To further quantify flows, we define the angle $\theta_c$ between the velocity of a cell and the $x$-axis aligned with the direction of the channel. $\langle \cos \theta_c \rangle$ averaged over cells and time is $\approx \pm 1$ for unidirectional flows, and $\approx 0$ if flows are predominantly chaotic. Alternatively, we also analyse the modified angle $\tilde{\theta}_c$ which is confined to the range $[0, \pi/2]$ so that $\tilde{\theta}_c = 0$ corresponds to the velocity in the direction along the channel, whereas $\tilde{\theta}_c = \pi/2$ corresponds to the velocity perpendicular to the channel. This additional metric would also detect shear flow profiles where cells in the bottom half of the channel move in the opposite direction to those in the top half, resulting in no net transport. Specifically, for unidirectional or shear flows, $\langle \tilde{\theta}_c \rangle \approx 0$, whereas for a chaotic flow, $\langle \tilde{\theta}_c \rangle \approx \pi/4$. Measurements of $\cos \theta_c$ and $\tilde{\theta}_c$, shown in Supplementary Figs. 2, 6, and 7, are calculated by averaging over all cells that are not attached to the wall and simulation snapshots between $t_1$ and $t_1 + t_2$ in steps $\delta t = 500$. They indeed agree with the unidirectional flow in the internal dissipation case and with the chaotic flow in the substrate dissipation case. Note that if MSD never exceeds $10^{-6}$, suggesting a fully arrested system, we assume $|\langle \cos(\theta_c) \rangle| = 0$ and $\langle \tilde{\theta}_c \rangle = \pi/4$.

### Correlation functions

For Fig. 5, we define the velocity-velocity correlation function as

$$C_{\mathbf{v}}(R) = \frac{\langle \mathbf{v}_c \cdot \mathbf{v}_{c'} \rangle_{R_{c,c'} \approx R}}{\langle \mathbf{v}_c \cdot \mathbf{v}_c \rangle_c}, \qquad (10)$$

where $\mathbf{v}_c$ is the velocity of cell $c$. The average in the numerator is over all pairs of cells $c$ and $c'$ whose centres are in the range $[R - \Delta R, R]$ (using the minimum-image convention[80]) and setting $\Delta R = 1$. The average in the denominator is over all cells in the tissue, hence, $C_{\mathbf{v}}(0) = 1$. Similarly, the director-director correlation function is given by

$$C_{\mathbf{Q}}(R) = \frac{\langle \mathbf{Q}_c : \mathbf{Q}_{c'} \rangle_{R_{c,c'} \approx R}}{\langle \mathbf{Q}_c : \mathbf{Q}_c \rangle_c}, \qquad (11)$$

where $\mathbf{Q}_c$ is the tensor used in determining the active force, and $:$ indicates full contraction. Note that the long correlations obtained necessitate system sizes and simulation times that are in general computationally too costly to study with the vertex model. Specifically, a system with $4 \times 10^4$ cells ($N_x = N_y = 200$) was simulated until

$t = 2 \times 10^4$. While these simulations are sufficient to establish that in the internal dissipation model, correlations are far longer than in the substrate dissipation case, it is possible that the correlations shown in Fig. 5 for the lower $\xi$ values have not yet fully developed and may have been affected by the finite size of the simulation box.

### Phase diagrams

To generate the phase diagrams in Fig. 4c and Supplementary Fig. 5, simulations are divided as follows. If MSD$(\tau = t_2/2)$ is less than 1, the parameter set is classified as non-motile. Otherwise, an $at^\beta$ fit is performed on that MSD data. For $\beta > 1.9$ the parameter set is classified as ballistic; otherwise, it is classified as sub-ballistic.

## Data availability

Source data are provided with this paper.

## Code availability

The vertex model code used in this study is based on an implementation initially developed by Matej Krajnc and is available from the corresponding author on request.

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

## Acknowledgements

We wish to thank M. Krajnc for providing the initial version of the vertex model code, and S. Bhattacharyya, S. Caballero-Mancebo, G. Charras, J. Klebs, A. Košmrlj, F. Mori, S. P. Thampi, A. Weber, and C.J. Weijer for many helpful discussions. J.R. and J.M.Y. acknowledge support from the UK Engineering and Physical Sciences Research Council (Award EP/W023849/1). J.M.Y. acknowledges support from the ERC Advanced Grant ActBio (funded as UKRI Frontier Research Grant EP/Y033981/1). K.V.S.C. and R.S. acknowledge support from the UK Engineering and Physical Sciences Research Council (Award EP/W023946/1).

## Author contributions

All authors designed the research. J.R. and K.V.S.C. performed and analysed numerical simulations under the guidance of J.M.Y. and R.S. The paper was written by J.R. with input from all other authors.

## Competing interests

The authors declare no competing interests.
