## [Transparent Peer Review file · Nature Communications]

Vertex model with internal dissipation enables sustained flows

Corresponding Author: Professor Rastko Sknepnek

Version 0:

Reviewer comments:

Reviewer #1

(Remarks to the Author)

The manuscript describes numerical simulations of a vertex model of cell epithelia focusing on two possible frictional terms in the equations of motion: a “dry friction” term represented as a drag force proportional to the local velocity and a “wet” friction force coupling the velocities of neighbouring cells. Simulations show that when wet friction is present, cells move parallel to each other along the direction of the channel, while dry friction leads to more diffusive and random flow.

I have several issues with the present manuscript relating to its originality with respect to the existing literature, the relevance vis a vis the experimental reality of epithelial motion and the depth of the analysis. In view of those concerns which I discuss in more details below, I think that the manuscript requires a very extensive revision and considerable additional work to be acceptable for publication.

While in vertex models one usually considers only a “dry” friction term, “wet” friction is a standard term in self-propelled particle models of active matter, starting from the pioneering papers of Vicsek, 30 years ago. The term was employed also when modelling the dynamics of epithelial cell sheets as for instance in Ref. 12 which is cited in the introduction but not mentioned when wet friction is discussed. An even more detailed study on the role of wet friction in driving directed motion in collective cell migration can be found in Chepizhko O, et al. *Soft matter*. 2018;14(19):3774-82 a paper that also includes a quantitative comparison with experiments.

This brings me to my second issue: the relevance of this study for the biological phenomena that the model is supposed to describe. The authors argue that dry friction is relevant for epithelial motion, while wet friction would instead rule collective motion in embryos. I disagree with this statement. In view of the previous literature on collective cell migration, I would conclude that both terms should be relevant in vitro as shown in the paper cited above. I would also like to call the authors attention to the paper by Iliina O et al. *Nature cell biology*. 2020;22(9):1103-15. In this paper, the authors report experiments in vitro where the transition between disordered/diffusive motion and directed is controlled by biological (ecadherin expression) and physical (collagen stiffness) controls. This demonstrates that both phases shown in the model can occur in the same system (epithelial-like sheets). According to the authors of the present manuscript, however, this would only happen when friction is switched from dry to wet (e.g. supposedly switching from in vitro experiments to in vivo embryos). The literature demonstrates instead that a switch from disordered to parallel flow is quite common in cell migration and can be triggered by multiple types of biological and physical controls.

Given that the novelty and applicability of the model is questionable, for the present manuscript to be a useful addition to the literature the authors should perform a much more thorough numerical analysis. At present, the authors only scratch the surface performing simulations in two extreme limits (e.g. only “dry” or only “wet” friction). One should instead study with much more care the crossover between these two regimes by varying the ratio between the two terms. I am also not convinced that parallel directed motion can not be induced even in absence of wet friction. I imagine that if the self-propulsion force is sufficiently strong the anisotropy provided by the channel will be enough to induce directed motion. There is no way to tell, since self-propulsion was not varied in this study. After decades of studies in active matter models, the minimal requirement for publication is to produce an in depth study where as many parameters as possible are varied and phase diagrams are drawn from simulations (see for instance the experimentally derived phase diagram in Iliina et). Here nothing of the sort is done and this is a serious shortcoming. In performing a deeper analysis, I also suggest to compute the local vorticity, its distribution and its profile. This will allow the authors to make contact with models of active turbulence

Reviewer #3

(Remarks to the Author)

In this manuscript, Rozman et al consider the nematic vertex model in two dimensions, which can be used as a model of confluent epithelial sheets. Prior investigations have considered the nematic vertex model in the “dry” limit, where dissipation in the model is due to friction with a substrate. Here, the authors extend this model to the “wet” limit, where the dissipation is internal, thereby extending the model to relevant regimes where the cells aren’t externally supported, such as early embryo development. By considering the model in a channel geometry, they demonstrate that the “wet” model generates sustained flows with long-range correlations present, contrasting the behavior of the “dry” model.

Overall, I believe these results provide an important step in developing computational models of cells that can describe biologically relevant regimes, and that the ideas presented here can be applied to future investigations of cellular behavior in regimes where the dissipation is internal.

There are some queries that I have, and points that should be addressed, prior to publication:

1) My main concern is whether the long-range correlations present in the system, as demonstrated in fig. 4, may be leading to finite-size effects in the smaller system sizes for which the main results are presented. Please verify whether this is the case. Also please explicitly state the system setup in terms of N_x and N_y for the system with 10^4 cells to give context on where the presented correlation results fit within the parameter space of the rest of the investigation.

2) When investigating the threshold activity for the onset of unidirectional flow, did you verify whether allowing the flow profile to develop without starting in the higher activity phase leads to the same mean velocity and activity threshold? Given how correlated the system is, I’m curious if correlations from the initial setup are sustained for long times.

Minor points:

- Please briefly justify the choice of a channel setup in the introduction and abstract.
- Lines 77-78, please use t and \hat{t} consistently.
- Line 88: please briefly justify the choice of $p_0=3.85$ and give context on where this value sits in the phase space of the model.
- Line 99: mention choice of N_x in the main text not just the figure caption.
- Lines 135-6: It’s unclear to me why one would expect the threshold activity to decay to zero when there is still friction present in the system. Please further justify this statement.
- Fig. 3 caption – panel a is not for $N_y=14$ only.
- Line 154: I believe should read “...and times of $\sim 10^5$ ”, based on the information elsewhere in the manuscript. Please verify.
- Lines 164 and Fig. S4 caption: please verify the value of p_0 used in the passive tissue.
- Lines 250-255: It’s not fully clear to me what $\langle \tilde{\theta}_c \rangle$ teaches us beyond what we learn from $\langle \cos(\theta_c) \rangle$.
- Please make it more explicit how the final paragraph of the methods, which notes how the model can behave like a solid at sufficiently low activity, ties in with your results and where this behavior may be being observed.
- Fig. S2: The points for $N=6$ aren’t visible. Please verify if they are hidden below other data points or are missing from the plot.
- Figs. S2 and S3 should be switched so that they are referenced in order.

Version 1:

Reviewer comments:

Reviewer #1

(Remarks to the Author)

My concerns about the present manuscript still remain after reading its revised version. The manuscript reports a model for active matter that represents a slight variation with respect to previously studied models. The connections with biological phenomena is not very strong since no experimental data are used to validate the model. I think that the manuscript could be more suited for a more specialized journal since it lacks general interest.

Reviewer #3

(Remarks to the Author)

The authors have addressed all of my comments. I am happy to recommend this manuscript for publication in Nature Communications.

Reviewer #4

(Remarks to the Author)

The editor asked to review the resubmitted manuscript and to review the exchange between the authors and reviewer #1.

Review of the manuscript:

In J. Rozman et al, the authors study the effects of internal and external dissipation on the spontaneous flow transition of active nematics using vertex models. First the authors introduce a generalization of the vertex model that includes internal dissipation as friction forces between neighboring vertices, external dissipation as friction forces between a vertex and a substrate and active stresses as an active force distributed on the vertices of each cell (with a vanishing monopole). Next the authors compare their simulations in a channel geometry to past analytical work on the spontaneous flow transition in active nematics. The authors vary the following parameters: the strength of the active stresses, the friction coefficient associated with internal dissipation, and the friction coefficient associated with external dissipation, the channel width, and the channel length. The authors identified several states in their simulations, in particular a non-flowing fluid state and a flowing fluid state. The authors characterize these states in a phase diagram as a function of the dimensionless active stress magnitude and the friction coefficient associated with external friction.

The manuscript is clear and well-written, the theoretical analysis is solid, and the main findings are novel. I have a few minor concerns that the authors may want to address regarding the interpretation of some of their results. Therefore, I recommend publication of the manuscript, when these minor comments are addressed.

Minor comments:

-In line 137-138 the authors state "For sufficiently high ζ the cell velocities reach a maximum in the centre of the channel and decrease towards the walls as in the continuum theories 50, 61-63". I believe that Ref 50 reported a simple shear flow profile rather than a Poiseuille flow profile. Can the authors correct it, if necessary. If the reference was incorrect, I would suggest the authors to check their citations, as the previous example may not be exhaustive.

-In paragraph 146, the authors report that in their simulations, the critical activity is independent on the channel width. Later, the authors argue that this effect might be because a force of fixed magnitude is sufficient to cause cell re-arrangements. I believe there could be another interpretation, which is that in Ref 50 it was studied the case of strong anchoring, whereas in this work, the authors seem to study a case of weak anchoring. In the limit of weak anchoring, it is known from analytical studies that the critical activity becomes independent on the channel width. Have the authors tested this possibility by imposing a preferred cell alignment on the boundaries?

-The authors state several times in the manuscript that in the dry limit, they do not find a spontaneous flowing state. From past analytical and numerical studies, it is expected that if the active stress magnitude is sufficiently large, the non-flowing phase can become unstable even in the presence of friction with the substrate. Can the authors explore values of the active coefficient beyond 0.2?

-Poiseuille-like vs shear-like flowing state: In Fig 2d, the authors validate their simulations by reproducing some past results of the spontaneous flow transition. In Ref. 50, it is predicted that the flowing state that is selected above threshold is a shear-flowing state. The corresponding velocity field is anti-symmetric with respect to a reflection on the channel midplane ($y=0$). In contrast the authors report a state that is symmetric with respect to a reflection on the channel midplane (Poiseuille-like flow). Can the authors explain this difference? Other past works like Ref. 61 and 63 reported that the flowing pattern above threshold is like a Poiseuille flow. Can the authors comment on this apparent contradiction with some past works?

Review of the exchange between reviewer #1 and the authors:

Overall, the authors have addressed satisfactorily most of the comments of reviewer #1. Reviewer #1 seem to be confused about the physics of this theoretical work. He/She considers that the authors were studying a system of self-propelled agents, whereas the authors studied a system of active nematic agents. This confusion gave rise to comments 1 and 5, and probably lead to the unjustified extra analysis that he/she requested in comment 6 and 7.

In comment 1, reviewer #1 questions the novelty of the results and claims that in simulations of self-propelled agents the effect of internal or external dissipation has already been studied. However, as the authors emphasize in their response,

their work focuses on simulations of active nematic agents. In a more general context, the authors study a class of active nematic (apolar) system, whereas the reviewer compare it to an active polar system. From a physical point of view, these are two different systems. For example, the flowing state reported in Fig 2 is generated by a different mechanism (see Ref. 50) than a flocking state. This point also addresses comment 5 of reviewer #1.

In comment 2-3, reviewer #1 question the relevance of their results for biological phenomena. The authors have amended the introduction and made significant text changes to address this comment. The resubmitted version of the manuscript better clarifies the biological relevance of internal vs external dissipation.

In comment 4, reviewer #1 ask for a more exhaustive search of the parameter space in the simulations. As the authors emphasize in their reply, reviewer #1 have missed several analyses in the original manuscript. Above, I listed parameters that were varied.

The extra analysis requested by Reviewer #1 in comments 6 and 7 are unjustified. There is no need to discuss comment 6. Regarding comment 7, I disagree about the relevance of studying the vorticity field because this theoretical work focuses on the spontaneous flow transition and the vorticity field provides analogous information to that reported in Fig 3.

Reviewer #5

(Remarks to the Author)

In this manuscript, Rozman et al. investigated how the cell-cell viscosity (referred to as internal dissipation by the authors) regulates the confined collective cell migration and the correlation length in two-dimensional epithelial cell sheets. By including the cell-cell viscosity, they extended the recently proposed active nematic vertex model where the cell activity is described as nematic (mimicking cellular active stresses, i.e., force dipoles) rather than polar (self-propulsion force). This kind of vertex model studied here is essential for studying tissue morphogenesis in the case without an underlying substrate or extracellular matrix, e.g., during the early stage of embryo development.

They explored the collective migration of epithelial cells confined in a channel by numerical simulations. They found that the cell-cell viscosity helps to establish a Poiseuille-like, unidirectional flow of collective cells, though the activity is nematic at the cellular level. This is because the cell-cell viscosity generates long-range correlations of cellular motions. They further explored how the results are affected by many parameters, including cell-cell adhesion, substrate friction, cellular activity, and channel confinement size.

This study is systematic and rigorous. The results are interesting and can be a good addition to studies of the collective cell dynamics during tissue morphogenesis.

The revised manuscript is well written. I have also read the comments from other referees and the responses by the authors. In my opinion, the authors have adequately addressed all the concerns raised by the referees.

Therefore, I recommend the publication of this revised manuscript. Before publication, I have some minor comments for the authors to address:

1. Reference update: Ref. 73 was recently published in PNAS (121, 37, e2405560121, 2024).
2. Typos in Figure S7: in panels (a,d), the title text in the figure should be " $p_0 = 1$ ".

Version 2:

Reviewer comments:

Reviewer #4

(Remarks to the Author)

The authors have addressed all my minor concerns, and I recommend publication of the present manuscript. Minor points asked several questions related to the behaviour of their vertex model near the instability of the ordered state. The authors have replied to these comments convincingly, made some additional analyses (Fig R1 and R2) and made some changes in the text of the main manuscript.

AUTHOR RESPONSE TO REVIEWER COMMENTS

Reviewer #1 (Remarks to the Author):

We thank the reviewer for providing feedback on our manuscript. We address the criticism raised by the reviewer below.

The manuscript describes numerical simulations of a vertex model of cell epithelia focusing on two possible frictional terms in the equations of motion: a “dry friction” term represented as a drag force proportional to the local velocity and a “wet” friction force coupling the velocities of neighbouring cells. Simulations show that when wet friction is present, cells move parallel to each other along the direction of the channel, while dry friction leads to more diffusive and random flow.

I have several issues with the present manuscript relating to its originality with respect to the existing literature, the relevance vis a vis the experimental reality of epithelial motion and the depth of the analysis. In view of those concerns which I discuss in more details below, I think that the manuscript requires a very extensive revision and considerable additional work to be acceptable for publication.

While in vertex models one usually considers only a “dry” friction term, “wet” friction is a standard term in self-propelled particle models of active matter, starting from the pioneering papers of Vicsek, 30 years ago. The term was employed also when modelling the dynamics of epithelial cell sheets as for instance in Ref. 12 which is cited in the introduction but not mentioned when wet friction is discussed. An even more detailed study on the role of wet friction in driving directed motion in collective cell migration can be found in Chepizhko O, et al. *Soft matter*. 2018;14(19):3774-82 a paper that also includes a quantitative comparison with experiments.

Response: We indeed did not sufficiently emphasise this in the original manuscript, but the activity studied in our work is not self-propulsion. The activity itself, rather than just the cell order, has nematic symmetry. Cells generate active stress that extends them along their long axis, but the net force a cell produces vanishes (i.e., the model allows only the presence of force dipoles).

As pointed out by the reviewer, self-propulsion models have been used extensively to understand a range of biologically relevant phenomena, starting from the pioneering work of Vicsek. However, as a direct consequence of Newton’s third law, cell self-propulsion requires that cells are in contact with an external structure, e.g. a substrate. In other words, for the cells to “walk/swim” in a direction, they need something to push or pull against. As such an external structure is not always present in systems where long-range correlated motion is observed (e.g., early avian embryos during gastrulation), it is important to understand how collective motion can also emerge without cell self-propulsion. Nematic activity, considered here, is the simplest activity allowed in such unsupported systems.

The choice of the “wet”/”dry” terminology was also not ideal: The terms are used in part of the active matter literature to classify active matter models into those where dissipation is momentum-conserving (called wet) and those where it is not (called dry). In self-propulsion models such as the works by Chepizhko, *et al.* mentioned by the Reviewer, a “ $(\mathbf{v}_i - \mathbf{v}_j)$ ” term is introduced to induce Vicsek-style aligning. However, as our model does not feature self-propulsion forces, the $(\mathbf{v}_i - \mathbf{v}_j)$ term takes only the role of internal dissipation, broadly corresponding to viscosity. This is also the reason why the $(\mathbf{v}_i - \mathbf{v}_j)$ model here is solved in the overdamped limit (i.e., at low Reynolds numbers), rather than the inertial case of the self-propulsion studies. Given the usual time- and length scales of tissue dynamics, the low Reynolds number regime is the relevant one for dissipation, but not necessarily for the dynamics of active self-propulsion velocities. In the revised manuscript, we make a comparison between the two models.

The idea of wet models is of course not new (in the original manuscript, when mentioning the concept of wet vs. dry dynamics, we cited the review of various wet vs. dry approaches; we have reworded the section in revision to clarify this). As pointed out by the Reviewer, they are indeed almost never considered

in vertex model studies. Vertex models are, however, a widely used class of models for understanding tissue dynamics. They encode physics that cannot be captured by other approaches (e.g., a vertex model features a density-independent rigidity transition at zero temperature that is not reproduced by equivalent particle models [Ref. 47 of the main text]) and allow for the tracking of the geometric properties of each cell. Given the unique physics of vertex models and their widespread application to a range of biological contexts, it is crucial to understand how they behave in the wet, rather than the usual dry limit, as the two encode very different physics.

As a Galilean-invariant model of dissipation, the “ $(\mathbf{v}_i - \mathbf{v}_j)$ ” approach would indeed fall under the heading of wet models, whereas the usual vertex-model approach of substrate-friction-dominated dynamics would fall under the dry models. However, that terminology is not widely accepted outside the field of active matter, so in the revised manuscript, we refer to the two models as the “internal dissipation” and “substrate dissipation”, respectively.

We have also modified the section of the manuscript discussing other mechanisms for collective motion - we referred to self-propulsion models there as polar, an alternative term used in part of the active matter literature, but less descriptive than self-propulsion.

This brings me to my second issue: the relevance of this study for the biological phenomena that the model is supposed to describe. The authors argue that dry friction is relevant for epithelial motion, while wet friction would instead rule collective motion in embryos. I disagree with this statement. In view of the previous literature on collective cell migration, I would conclude that both terms should be relevant *in vitro* as shown in the paper cited above. I would also like to call the authors attention to the paper by Iilina O et al. Nature cell biology. 2020;22(9):1103-15. In this paper, the authors report experiments *in vitro* where the transition between disordered/diffusive motion and directed is controlled by biological (ecadherin expression) and physical (collagen stiffness) controls. This demonstrates that both phases shown in the model can occur in the same system (epithelial-like sheets). According to the authors of the present manuscript, however, this would only happen when friction is switched from dry to wet (e.g. supposedly switching from *in vitro* experiments to *in vivo* embryos).

Response: The introduction of the original manuscript was, unfortunately, not sufficiently precise and gave the wrong impression as to our claims; we have substantially rewritten it in the revision. To summarise, we aimed to draw attention to the fact that two modes of dissipation can exist in tissues: internal dissipation and friction with the substrate (or some other external structure). While internal dissipation should to some extent be present in all confluent tissues, substrate friction, being non-Galilean-invariant, inherently requires some sort of external structure. Early embryos, such as the chicken during gastrulation, were mentioned as examples of biological systems where dissipation can only be internal due to the absence of an external structure, making the widespread omission of such terms in confluent-tissue modelling problematic. However, we in no way make the reverse claim: In general, cells should experience both internal and substrate dissipation when in contact with some external structure that allows for the latter. Therefore, modelling internal dissipation should be of relevance to all confluent tissues, *in vivo* and *in vitro*, but substrate dissipation is not always applicable.

The literature demonstrates instead that a switch from disordered to parallel flow is quite common in cell migration and can be triggered by multiple types of biological and physical controls.

Response: As is shown in the paper, the dissipation mode is of course not the only control parameter for the transition to collective motion. Biological properties of the system, such as cell-cell adhesion and junctional tension, expressed through the target perimeter p_0 (Figs. S4, S6, S7) and the extent of cell activity (Figs. 2f,g, 3a, 4, S2, S4, S5-S7), as well as physical controls such as the substrate friction (Figs. 2e-g, 4, S2, S5-S7) all control the type of motion observed.

Rather, the manuscript states that in cell-level active nematic systems, internal dissipation is a necessary, but not a sufficient condition for sustained collective motion. In the revised manuscript, we draw

attention to the relevant experimental literature, including the work by Ilina O, *et al.*, Nature cell biology. 2020;22(9):1103-15, when discussing how the various parameters in the model control the transition to sustained flows.

Given that the novelty and applicability of the model is questionable, for the present manuscript to be a useful addition to the literature the authors should perform a much more thorough numerical analysis. At present, the authors only scratch the surface performing simulations in two extreme limits (e.g. only “dry” or only “wet” friction). One should instead study with much more care the crossover between these two regimes by varying the ratio between the two terms.

Response: We respectfully disagree with the Reviewer’s comments here and in the following two questions: Detailed parameter sweeps for the majority of the parameters have been carried out in the original manuscript, but were mostly shown in Supplemental Material to aid the flow of the manuscript. In the revised manuscript, we have substantially expanded these with additional simulation runs and quantifications and we have partially moved them to the main text to clarify what simulations were performed to substantiate the claims of the paper.

Specifically, for the question of how the ratio of the substrate and internal friction impacts the dynamics, we augmented the original Fig. 2f, showing the changes in velocity along the channel, and Fig. 4, showing changes in correlation (now Figs. 4a and 5, respectively) by also analysing how the type of transport (Fig. 4b) changes as substrate friction is increased from the internal dissipation limit in Fig. 2a through the transition to chaotic flows. This data was then organised into phase diagrams in Fig. 4c and Supplementary Fig. 5. Note that as the internal dissipation coefficient η is set to 1 in the non-dimensionalisation procedure to define the time-scale of the internal dissipation model, it is sufficient to only vary the substrate friction coefficient ξ without loss of generality.

I am also not convinced that parallel directed motion can not be induced even in absence of wet friction. I imagine that if the self-propulsion force is sufficiently strong the anisotropy provided by the channel will be enough to induce directed motion. There is no way to tell, since self-propulsion was not varied in this study.

Response: As noted above, the model does not feature a self-propulsion term and the activity driving flows is nematic. In the original manuscript, we varied this activity over a range of values and for a number of channel widths, finding unidirectional flows only if internal dissipation is included, but the results were shown in the Supplemental Material (original Fig. S3). In the revised manuscript, we have partially moved these plots into the main text (Fig. 2f,g) to clarify the simulations substantiating the claims of the paper. We have also expanded the analysis, performing additional simulations far in the solid and fluid regimes of the passive model (Figs. S6, S7), confirming that while the internal dissipation model generically flows in the active nematic fluid phase, the substrate dissipation model does not.

In the original manuscript, we did not sufficiently clarify the physical underpinnings of the transition. In the revised manuscript, we have extended the discussion concluding the Results section highlighting why internal dissipation is crucial for unidirectional motion. To summarise, for unidirectional motion down a channel to emerge, correlation lengths in the systems have to be comparable with channel width. As the model tissue is fluid-like, long-range correlations should only be able to arise through hydrodynamic effects (i.e., through viscosity/internal dissipation). Inversely, substrate friction in general screens hydrodynamics-induced correlation and reduces the correlation length. Therefore, motion is chaotic in the cases that consider only substrate friction (or else the substrate friction is dominant over internal dissipation), as the correlation length is less than the width of the channel. Introducing internal dissipation allows for longer correlation lengths to emerge, leading to a transition to channel-wide flows. This explains why the transition to unidirectional flows cannot be induced by only increasing the activity in the substrate dissipation model: The key problem preventing unidirectional flow with only substrate friction is the absence of a mechanism for the necessary long-range velocity correlations.

Lastly, we stress that the analysed range of activities covers the physically relevant set of parameters. The lowest activities no longer correspond to flows, whereas the highest ones correspond to very elongated cells (cell shape index exceeds 5.3) suggesting further increases in the activity would take the model far from realistic cell shapes for epithelia.

After decades of studies in active matter models, the minimal requirement for publication is to produce an in depth study where as many parameters as possible are varied and phase diagrams are drawn from simulations (see for instance the experimentally derived phase diagram in Ilina et). Here nothing of the sort is done and this is a serious shortcoming.

Response: In the revised manuscript, we have clarified that almost all parameters of the model (friction, activity, target perimeter, channel width, and channel length) were varied (Figs. 2f,g, 3a,b, 4, 5, S2, S4-S8). The key transition, spanning friction and activity, was arranged into phase diagrams in Figs. 4c and S5.

In performing a deeper analysis, I also suggest to compute the local vorticity, its distribution and its profile. This will allow the authors to make contact with models of active turbulence (Alert R et al. Annual Review of Condensed Matter Physics. 2022 Mar 10;13:143-70.).

Response: We thank the Reviewer for suggesting analysing the vorticity and we have done so in preparing the revision. The results obtained indeed show a transition from unidirectional flow at low substrate friction to chaotic motion at high substrate friction in channel confinement. However, as we are principally interested in the transition to collective motion and to keep the paper concise, we rather base the revised manuscript on the MSD analysis as it gives a more direct and accurate measure of that transition. For completeness, the vorticity analysis is shown here in Fig. R1 below.

Figure R1. a-d) Vorticity profile in the channel for different levels of substrate friction, showing the transition from unidirectional to chaotic flows: $\xi = 10^{-5}$ (a), $\xi = 10^{-3}$ (b), $\xi = 10^{-2}$ (c), and $\xi = 10^{-1}$ (d). e) Vorticity-vorticity correlations in a channel for different substrate frictions. f) Vorticity-vorticity correlations in a periodic tissue for different substrate frictions. All panels are for $\zeta = 0.1$ and $\eta = 1$; channels have $N_x = 14$.

Reviewer #3 (Remarks to the Author):

In this manuscript, Rozman et al consider the nematic vertex model in two dimensions, which can be used as a model of confluent epithelial sheets. Prior investigations have considered the nematic vertex model in the “dry” limit, where dissipation in the model is due to friction with a substrate. Here, the authors extend this model to the “wet” limit, where the dissipation is internal, thereby extending the model to relevant regimes where the cells aren’t externally supported, such as early embryo development. By considering the model in a channel geometry, they demonstrate that the “wet” model generates sustained flows with long-range correlations present, contrasting the behavior of the “dry” model.

Overall, I believe these results provide an important step in developing computational models of cells that can describe biologically relevant regimes, and that the ideas presented here can be applied to future investigations of cellular behaviour in regimes where the dissipation is internal.

We thank the reviewer for their critical reading of our manuscript and an overall positive assessment of it.

There are some queries that I have, and points that should be addressed, prior to publication:

1) My main concern is whether the long-range correlations present in the system, as demonstrated in fig. 4, may be leading to finite-size effects in the smaller system sizes for which the main results are presented. Please verify whether this is the case. Also please explicitly state the system setup in terms of N_x and N_y for the system with 10^4 cells to give context on where the presented correlation results fit within the parameter space of the rest of the investigation.

Response: In the revised manuscript, we compare the flow profiles that emerge in the currently studied channels at length $N_x = 48$ with those in a channel with length $N_x = 200$ as well as how the mean velocities along the channel depend on activity for these two channel lengths (Supplementary Fig. 8). The results are essentially identical for both lengths, suggesting that our choice of N_x does not significantly affect the results. We also clarify that the periodic simulations are done for a $N_x = N_y = 200$ cell system.

2) When investigating the threshold activity for the onset of unidirectional flow, did you verify whether allowing the flow profile to develop without starting in the higher activity phase leads to the same mean velocity and activity threshold? Given how correlated the system is, I’m curious if correlations from the initial setup are sustained for long times.

Response: We did not clarify this sufficiently in the original manuscript: The procedure in which we start from high activities so that the flow profile develops, and then reduce it to a final value is only used for the inset of Fig. 3a. For the main plot in Fig. 3a and 3b, simulations have a constant activity throughout. There is indeed a small difference in threshold activities so that a slightly smaller activity can sustain flows than is required for them to spontaneously develop, at least on the time scale of the simulation. Highlighting this was the original reason for including the inset of 3a, and we have clarified this point in the revised manuscript. Furthermore, we have now added Supplementary Fig. 3, which shows that aside from the mentioned behaviour near the threshold value, mean velocities are effectively the same if the system starts from a high activity or from the final one.

Minor points: - Please briefly justify the choice of a channel setup in the introduction and abstract.

Response: In the revised manuscript and abstract, we clarify that we are focusing on channels as active nematics confined to a channel have been shown to produce unidirectional flows both theoretically and experimentally. As we are interested in the minimal conditions for unidirectional motion, this is a convenient geometry to study the dynamics of our model.

- Lines 77-78, please use t and \hat{t} consistently.

Response: We thank the reviewer for pointing this out and have corrected it.

- Line 88: please briefly justify the choice of $p_0 = 3.85$ and give context on where this value sits in the phase space of the model.

Response: In the revised manuscript, we clarify that $p_0 = 3.85$ corresponds to the passive vertex model being slightly above the initially reported rigidity transition threshold at 3.81. In the revised manuscript, we also add detailed analysis for $p_0 = 1$ (far in the solid regime) and $p_0 = 4$ (well above the reported range of values for the passive rigidity transition) (Supplementary Figs. 6 and 7).

- Line 99: mention choice of N_x in the main text not just the figure caption.

Response: We now state that the channel is $N_x = 48$ cells long.

- Lines 135-6: It's unclear to me why one would expect the threshold activity to decay to zero when there is still friction present in the system. Please further justify this statement.

Response: We wished to make it explicit that the absence of flows at low activities is not simply due to the rigidity transition in the passive vertex model, so we stressed that they do not develop even for $p_0 = 4$, which is above all reported values of the rigidity transition threshold (there is some ambiguity as to the value of the rigidity transition in the literature). We now make this point more clearly so as not to give the impression a 0 threshold activity is expected.

- Fig. 3 caption – panel a is not for $N_y = 14$ only.

Response: We have corrected the caption in the revised manuscript.

- Line 154: I believe should read “...and times of $\sim 10^5$ ”, based on the information elsewhere in the manuscript. Please verify.

Response: We phrased the section around line 154 poorly: We wanted to draw a comparison with standard times and sizes in the vertex model literature in general, rather than contrast with the rest of our manuscript. We agree that it was unclear and have removed the comparison in the revised manuscript.

- Lines 164 and Fig. S4 caption: please verify the value of p_0 used in the passive tissue.

Response: The target perimeter of the passive tissue is intentionally selected to be far in the passive region, so we set $p_0 = 1$. This is clarified in the revised manuscript.

- Lines 250-255: It's not fully clear to me what $\langle \tilde{\theta}_c \rangle$ teaches us beyond what we learn from $\langle \cos(\theta_c) \rangle$.

Response: As a metric of channel flows, $\langle \cos(\theta_c) \rangle$ would capture unidirectional flows, but not shear flows in which cells in the bottom half move in the opposite direction to those in the top half, resulting in channel-wide flows but no net transport. We, therefore, added $\langle \tilde{\theta}_c \rangle$, which is confined to the range $[0, \pi/2]$ so that it also captures shear flow profiles. We clarify this distinction in the revised manuscript.

- Please make it more explicit how the final paragraph of the methods, which notes how the model can behave like a solid at sufficiently low activity, ties in with your results and where this behaviour may be being observed.

Response: In the revised manuscript, we added a section discussing how the model behaves for different target perimeters p_0 (Supplementary Figs. 6 and 7). Specifically, if p_0 is below the passive rigidity transition value ~ 3.81 , the model only behaves as an active nematic fluid if the activity is above a threshold value ζ_* . In this active nematic state, internal dissipation is required for ballistic transport down the channel to emerge for all studied p_0 . For a range of activities below ζ_* , the tissue is instead in a “rhombile” state, which has a finite shear modulus. Therefore, as the rhombile state is solid-like, internal dissipation is not necessary for long-range correlations. We indeed find that ballistic transport can develop with only substrate friction in the rhombile state. However, this state does not resemble commonly studied epithelia, so this regime is of limited interest.

- Fig. S2: The points for $N=6$ aren't visible. Please verify if they are hidden below other data points

or are missing from the plot.

Response: The $N = 6$ points were hidden under other data. In the revised manuscript, we have modified the plot so that each N uses a different marker and all values can be seen.

- Figs. S2 and S3 should be switched so that they are referenced in order.

Response: We thank the reviewer for pointing this out, and we have corrected it in the revised manuscript.

AUTHOR RESPONSE TO REVIEWERS' COMMENTS

Reviewer #1 (Remarks to the Author):

My concerns about the present manuscript still remain after reading its revised version. The manuscript reports a model for active matter that represents a slight variation with respect to previously studied models. The connections with biological phenomena is not very strong since no experimental data are used to validate the model. I think that the manuscript could be more suited for a more specialized journal since it lacks general interest.

We thank the reviewer for taking the time to read the revised manuscript. We are disappointed that they still do not find our work of general interest.

Reviewer #3 (Remarks to the Author):

The authors have addressed all of my comments. I am happy to recommend this manuscript for publication in Nature Communications.

We thank the reviewer for their rereading of the manuscript and their supportive assessment.

Reviewer #4 (Remarks to the Author):

The editor asked to review the resubmitted manuscript and to review the exchange between the authors and reviewer #1.

Review of the manuscript:

In J. Rozman et al, the authors study the effects of internal and external dissipation on the spontaneous flow transition of active nematics using vertex models. First the authors introduce a generalization of the vertex model that includes internal dissipation as friction forces between neighboring vertices, external dissipation as friction forces between a vertex and a substrate and active stresses as an active force distributed on the vertices of each cell (with a vanishing monopole). Next the authors compare their simulations in a channel geometry to past analytical work on the spontaneous flow transition in active nematics. The authors vary the following parameters: the strength of the active stresses, the friction coefficient associated with internal dissipation, and the friction coefficient associated with external dissipation, the channel width, and the channel length. The authors identified several states in their simulations, in particular a non-flowing fluid state and a flowing fluid state. The authors characterize these states in a phase diagram as a function of the dimensionless active stress magnitude and the friction coefficient associated with external friction.

The manuscript is clear and well-written, the theoretical analysis is solid, and the main findings are novel. I have a few minor concerns that the authors may want to address regarding the interpretation of some of their results. Therefore, I recommend publication of the manuscript, when these minor comments are addressed.

We appreciate the reviewer's support as well as the points raised, which are addressed below.

Minor comments:

-In line 137-138 the authors state “For sufficiently high ζ the cell velocities reach a maximum in the centre of the channel and decrease towards the walls as in the continuum theories 50, 61-63”. I believe that Ref 50 reported a simple shear flow profile rather than a Poiseuille flow profile. Can the authors correct it, if necessary. If the reference was incorrect, I would suggest the authors to check their citations, as the previous example may not be exhaustive.

We thank the Reviewer for pointing this out. Ref. 50 was indeed not appropriate here. In the revised manuscript, we modified the reference to D. Marenduzzo et al. Phys. Rev. E 76, 031921 (2007), which reports a Poiseuille-like flow of an active nematic in a channel (please also see the response below regarding the Poiseuille vs shear flow).

-In paragraph 146, the authors report that in their simulations, the critical activity is independent on the channel width. Later, the authors argue that this effect might be because a force of fixed magnitude is sufficient to cause cell re-arrangements. I believe there could be another interpretation, which is that in Ref 50 it was studied the case of strong anchoring, whereas in this work, the authors seem to study a case of weak anchoring. In the limit of weak anchoring, it is known from analytical studies that the critical activity becomes independent on the channel width. Have the authors tested this possibility by imposing a preferred cell alignment on the boundaries?

We thank the reviewer for this insightful comment. The section that made a comparison with the results in Ref. 50 could have been worded better. While Ref. 50 analysed a transition from a well-aligned to a flowing state, the transition reported in our work is fundamentally different. As activity increases, flows first develop in the ‘rhombile’ state with plug-like flows (Fig. 3b inset and Fig. 3c in the manuscript). This flow transitions to the active nematic state with Poiseuille-like flows following a small further increase in the activity. Given that the initial transition to flows occurs in a regime where the model does not resemble active nematics and is not well aligned, predictions of pure active nematic theories for the onset of flows are likely not applicable. In the revised manuscript, we emphasise this point.

Introducing explicit anchoring terms to the model is complicated, as the \mathbf{Q} field is not an independent variable but a readout of cell shape which therefore cannot be easily independently set. This makes it difficult to compare directly with the different anchoring conditions discussed in the literature. We agree that our explanation for why the low-activity plug flow thresholds are width-independent was not sufficiently justified, and we have removed it in the revised manuscript.

-The authors state several times in the manuscript that in the dry limit, they do not find a spontaneous flowing state. From past analytical and numerical studies, it is expected that if the active stress magnitude is sufficiently large, the non-flowing phase can become unstable even in the presence of friction with the substrate. Can the authors explore values of the active coefficient beyond 0.2?

Increasing the activity in the model produces increasingly elongated cells. At $\zeta = 0.2$, the shape index (perimeter over square root of area) already exceeds 5.3, which is very high for the vertex model cells that assume straight cell-cell junctions. We therefore limit the analysis in the paper to $\zeta \leq 0.2$ to remain within the domain of realistic epithelial cell shapes. However, increasing the activity to 0.45 (near the region where simulations become unstable) shows that the substrate dissipation model remains approximately diffusive and does not transition to ballistic motion/flows along the channel (Fig. R1).

Figure R1: (a-c) Analysis of cell motion in the channel for substrate dissipation dynamics at higher activities: Exponent of MSD power-law fit (a), absolute value of the average cosine of the angle between cell velocity and the x -axis (b), and average of that angle confined to the range $[0, \pi/2]$ (c). (d,e) Model tissue with substrate dissipation dynamics at $\zeta = 0.2$ (d) and 0.45 (e).

Furthermore, the difference between the substrate dissipation and internal dissipation case is not between a non-moving and a flowing tissue. Rather, activity can still fluidise the tissue in the presence of only substrate friction, causing cells to become motile (e.g. Fig. 2 c,e in the manuscript). However, because there is no internal dissipation (i.e. no viscosity), there is no physical mechanism for long-range correlations to emerge (Fig. 5 in the manuscript). As unidirectional flows along the channel would require the correlation lengths to be comparable with channel width (with the substrate dissipation alone correlations are of order ~ 1 cell [Fig. 5 in the manuscript]), the activity-fluidised tissue immediately shows active-turbulence-like chaotic flows, rather than system-wide organised motion. To the best of our knowledge, instead of fully omitting viscosity, past continuum studies incorporated friction while retaining viscosity, thereby preserving a correlation mechanism. This leads to the difference in predicted behaviour compared to the results obtained in the vertex model when only friction is considered.

-Poiseuille-like vs shear-like flowing state: In Fig 2d, the authors validate their simulations by reproducing some past results of the spontaneous flow transition. In Ref. 50, it is predicted that the flowing state that is selected above threshold is a shear-flowing state. The corresponding velocity field is anti-symmetric with respect to a reflection on the channel midplane ($y=0$). In contrast the authors report a state that is symmetric with respect to a reflection on the channel midplane (Poiseuille-like flow). Can the authors explain this difference? Other past works like Ref. 61 and 63 reported that the flowing pattern above threshold is like a Poiseuille flow. Can the authors comment on this apparent contradiction with some past works?

We thank the reviewer for raising this important point. As noted by the reviewer, R. Voituriez et al. *Europhys. Lett.* 70, 404 (2005) reported a spontaneous transition to shear flow with a node (zero fluid velocity) at the channel centerline via stability analysis. However, D. Marenduzzo, et al. *Phys. Rev. E* 76, 031921 (2007), used full numerical simulations to demonstrate that the shear flow predicted by Voituriez et al. is a metastable state. They reported that near the transition threshold, the shear flow could persist for a long time, with the system eventually transitioning to a steady-state Poiseuille flow, with a maximum flow velocity—rather than a nodal point—at the centre of the channel. Additional studies [e.g. S. Chandragiri et al., *Soft Matter* 15, 1597 (2019), and A. Samui et al., *Soft Matter* 17, 10640 (2021)] reported a direct transition to unidirectional Poiseuille flow without any intermediate shear flow. In contrast, G. Duclos et al., *Nat. Phys.* 14, 728 (2018) reported a steady-state shear flow in channel-confined retinal pigment epithelial cells, supported by numerical simulations based on continuum active nematic theory. Selection of the state is likely to be complex as these are non-equilibrium steady states. Therefore, which state is achieved will depend on the initial conditions and system history. We are unaware of any literature that definitively clarifies the factors governing the selection between these two distinct flow states.

Therefore, we just remark that the two flow profiles are also associated with different nematic textures; in Poiseuille flow, the director's texture forms a nozzle-like structure converging toward the centerline, while in shear flow, the director maintains a uniform, preferred orientation across the channel. The texture obtained in our simulations is consistent with that of a Poiseuille flow (Fig. R2). Lastly, in the revised manuscript, we now emphasise that the flow profile is Poiseuille-like.

Figure R2: Panels (a) and (b) show the nematic textures from the continuum model [S. Chandragiri et al. (2019)] and the current study, respectively. Both correspond to a Poiseuille flow, where the director texture appears to converge toward the centerline, forming a nozzle-like structure. In contrast, panel (c) shows the director texture corresponding to shear flow from the continuum model, as reported by G. Duclos et al. (2018), where the director maintains a uniform orientation across the channel.

Review of the exchange between reviewer #1 and the authors:

Overall, the authors have addressed satisfactorily most of the comments of reviewer #1. Reviewer #1 seem to be confused about the physics of this theoretical work. He/She considers that the authors were studying a system of self-propelled agents, whereas the authors studied a system of active nematic agents. This confusion gave rise to comments 1 and 5, and probably lead to the unjustified extra analysis that he/she requested in comment 6 and 7.

In comment 1, reviewer #1 questions the novelty of the results and claims that in simulations of self-propelled agents the effect of internal or external dissipation has already been studied. However, as the authors emphasize in their response, their work focuses on simulations of active nematic agents. In a more general context, the authors study a class of active nematic (apolar) system, whereas the reviewer compare it to an active polar system. From a physical point of view, these are two different

systems. For example, the flowing state reported in Fig 2 is generated by a different mechanism (see Ref. 50) than a flocking state. This point also addresses comment 5 of reviewer #1.

In comment 2-3, reviewer #1 question the relevance of their results for biological phenomena. The authors have amended the introduction and made significant text changes to address this comment. The resubmitted version of the manuscript better clarifies the biological relevance of internal vs external dissipation.

In comment 4, reviewer #1 ask for a more exhaustive search of the parameter space in the simulations. As the authors emphasize in their reply, reviewer #1 have missed several analyses in the original manuscript. Above, I listed parameters that were varied.

The extra analysis requested by Reviewer #1 in comments 6 and 7 are unjustified. There is no need to discuss comment 6. Regarding comment 7, I disagree about the relevance of studying the vorticity field because this theoretical work focuses on the spontaneous flow transition and the vorticity field provides analogous information to that reported in Fig 3.

Reviewer #5 (Remarks to the Author):

In this manuscript, Rozman et al. investigated how the cell-cell viscosity (referred to as internal dissipation by the authors) regulates the confined collective cell migration and the correlation length in two-dimensional epithelial cell sheets. By including the cell-cell viscosity, they extended the recently proposed active nematic vertex model where the cell activity is described as nematic (mimicking cellular active stresses, i.e., force dipoles) rather than polar (self-propulsion force). This kind of vertex model studied here is essential for studying tissue morphogenesis in the case without an underlying substrate or extracellular matrix, e.g., during the early stage of embryo development.

They explored the collective migration of epithelial cells confined in a channel by numerical simulations. They found that the cell-cell viscosity helps to establish a Poiseuille-like, unidirectional flow of collective cells, though the activity is nematic at the cellular level. This is because the cell-cell viscosity generates long-range correlations of cellular motions. They further explored how the results are affected by many parameters, including cell-cell adhesion, substrate friction, cellular activity, and channel confinement size.

This study is systematic and rigorous. The results are interesting and can be a good addition to studies of the collective cell dynamics during tissue morphogenesis.

The revised manuscript is well written. I have also read the comments from other referees and the responses by the authors. In my opinion, the authors have adequately addressed all the concerns raised by the referees.

Therefore, I recommend the publication of this revised manuscript. Before publication, I have some minor comments for the authors to address:

We thank the Reviewer for their careful reading of their manuscript and their supportive comments.

1. Reference update: Ref. 73 was recently published in PNAS (121, 37, e2405560121, 2024).

Thank you. We have updated reference 73, and Ref. 69, which has also recently been published.

2. Typos in Figure S7: in panels (a,d), the title text in the figure should be “ $p_0 = 1$ ”.

Thank you. We have corrected this.

AUTHOR RESPONSE TO REVIEWERS' COMMENTS

Reviewer #4 (Remarks to the Author):

The authors have addressed all my minor concerns, and I recommend publication of the present manuscript. Minor points asked several questions related to the behaviour of their vertex model near the instability of the ordered state. The authors have replied to these comments convincingly, made some additional analyses (Fig R1 and R2) and made some changes in the text of the main manuscript.

We appreciate very much the reviewer's supportive assessment.